# Current and New Challenges in the Management of Pancreatic Neuroendocrine Tumors: The Role of miRNA-Based Approaches as New Reliable Biomarkers

**DOI:** 10.3390/ijms23031109

**Published:** 2022-01-20

**Authors:** Andrei Havasi, Daniel Sur, Simona Sorana Cainap, Cristian-Virgil Lungulescu, Laura-Ioana Gavrilas, Calin Cainap, Catalin Vlad, Ovidiu Balacescu

**Affiliations:** 1Department of Medical Oncology, The Oncology Institute “Prof. Dr. Ion Chiricuta”, 400015 Cluj-Napoca, Romania; havasi.andrei@gmail.com (A.H.); calincainap2015@gmail.com (C.C.); 211th Department of Medical Oncology, University of Medicine and Pharmacy “Iuliu Hatieganu”, 400015 Cluj-Napoca, Romania; obalacescu@yahoo.com; 3MedEuropa Radiotherapy Center, 410191 Oradea, Romania; 4Department of Mother and Child, Pediatric Cardiology, University of Medicine and Pharmacy “Iuliu Hatieganu”, 400015 Cluj-Napoca, Romania; cainap.simona@umfcluj.ro; 5Department of Medical Oncology, University of Medicine and Pharmacy Craiova, 200349 Craiova, Romania; cristilungulescu@yahoo.com; 6Department of Bromatology, Hygiene, Nutrition, University of Medicine and Pharmacy “Iuliu Hatieganu”, 23 Marinescu Street, 400337 Cluj-Napoca, Romania; laura.biris@umfcluj.ro; 7Department of Surgery, The Oncology Institute “Prof. Dr. Ion Chiricuta”, 34–36, Republicii Street, 400015 Cluj-Napoca, Romania; catalinvlad@yahoo.it; 8Department of Oncology, “Iuliu Hatieganu” University of Medicine and Pharmacy, 8, Victor Babes Street, 400012 Cluj-Napoca, Romania; 9Department of Genetics, Genomics and Experimental Pathology, The Oncology Institute “Prof. Dr. Ion Chiricuta’’, 400015 Cluj-Napoca, Romania

**Keywords:** MicroRNAs, pancreatic neuroendocrine tumors (PanNETs), biomarkers, diagnosis, prognosis, therapeutic, genetic

## Abstract

Pancreatic neuroendocrine tumors (PanNETs) are rare tumors; however, their incidence greatly increases with age, and they occur more frequently among the elderly. They represent 5% of all pancreatic tumors, and despite the fact that low-grade tumors often have an indolent evolution, they portend a poor prognosis in an advanced stages and undifferentiated tumors. Additionally, functional pancreatic neuroendocrine tumors greatly impact quality of life due to the various clinical syndromes that result from abnormal hormonal secretion. With limited therapeutic and diagnostic options, patient stratification and selection of optimal therapeutic strategies should be the main focus. Modest improvements in the management of pancreatic neuroendocrine tumors have been achieved in the last years. Therefore, it is imperative to find new biomarkers and therapeutic strategies to improve patient survival and quality of life, limiting the disease burden. MicroRNAs (miRNAs) are small endogenous molecules that modulate the expression of thousands of genes and control numerous critical processes involved in tumor development and progression. New data also suggest the implication of miRNAs in treatment resistance and their potential as prognostic or diagnostic biomarkers and therapeutic targets. In this review, we discusses the current and new challenges in the management of PanNETs, including genetic and epigenetic approaches. Furthermore, we summarize the available data on miRNAs as potential prognostic, predictive, or diagnostic biomarkers and discuss their function as future therapeutic targets.

## 1. Introduction

Pancreatic neuroendocrine tumors (PanNETs) are a rare heterogeneous group of tumors that develop from the neuroendocrine cells of the pancreas, with an incidence of <1 per 100,000 population. However, the incidence increases with age, and in the seventh decade of life, the incidence rates increase to 16 cases per 100,000 population [1]. Tumor grade, Ki-67 proliferation index, and differentiation are fundamental for the classification of PanNETs and to inform therapeutic decision-making, as they are strongly related to outcome. The World Health Organization classification divides pancreatic neuroendocrine neoplasms into well-differentiated PanNETs, poorly differentiated neuroendocrine carcinoma (PanNEC), and mixed neuroendocrine and non-neuroendocrine neoplasms [2]. PanNETs may be further classified as functional or nonfunctional, based on the presence of inappropriate hormone production and corresponding clinical syndromes. Up to 90% of all PanNETs are nonfunctional or secrete low levels of hormones, not enough to produce symptoms or secrete hormones without clinical manifestations. More than half of patients present with localized disease, and up to 27% of cases have metastatic disease at presentation [3]. The clinical syndromes derived from ectopic hormonal secretion lead to an earlier diagnosis for patients with functional PanNETs and better survival than those with nonfunctional tumors [4]. PanNETs are also associated with a substantial economic burden. A Swedish population-registry-based study observed an annual cost of EUR 37,000 per patient with low-grade metastatic PanNETs, similar to breast and prostate cancer [5].

Since their discovery, microRNAs (miRNAs) have been considered master modulators of gene expression, and unique miRNA signatures were linked to various genetic, metabolic, infectious, and autoimmune diseases [6]. Moreover, overwhelming data have associated MiRNA alteration with tumor development and progression, suggesting their potential as diagnostic, prognostic, and therapeutic biomarkers in several human cancers [7]. However, there is little data available concerning the involvement of microRNAs in PanNETs. Therefore, in this review, we will summarize the current diagnostic and therapeutic strategies available for the management of PanNETs and discuss the main genetic and epigenetic alterations associated with PanNETs, focusing on the involvement of miRNAs in pathogenesis and their role as emerging biomarkers.

## 2. Current Diagnostic and Therapeutic Strategies in PanNETs

### 2.1. Diagnostic

The diagnostic sequence varies, depending on the presentation, and encompasses an array of imagistic studies, serological markers, and, ultimately, immunohistochemical examination of the tumor tissue for confirmation. Accurate imaging is essential for both diagnosis and treatment selection. Initial imaging studies typically consist of contrast-enhanced computer tomography, as it is widely available and provides good visualization of the tumor and possible lymphatic or distant metastases, with sensitivity and specificity varying from 63–82% to 83–100% for tumors bigger than 2 cm. Magnetic resonance imaging has superior sensitivity and specificity and is considered a second-line imagistic investigation generally reserved for detecting tumors smaller than 2 cm or small liver metastases. Endoscopic ultrasound is suitable for the detection of tumors smaller than 2 cm, and it allows for fine-needle aspirations. However, it is limited by the endoscopist experience and has limited value in evaluating the extent of metastatic disease [8].

Somatostatin receptors are expressed in 80–100% of both functional and non-functional, well-differentiated PanNETs, except for insulinomas, for which the expression rate is lower (50–70%) compared to other subtypes. Poorly differentiated tumors do not express or have low somatostatin-receptor expression [9,10]. Functional imaging using radiolabeled somatostatin analogs with ^111^In-pentetreotide single-photon emission computed tomography and ^68^Ga-DOTA-SSA positron emission computed tomography (PET/CT) allows for tumor detection and accurate assessment of disease burden with higher sensitivity and specificity than conventional imaging. Furthermore, functional imaging is essential for well-differentiated tumors with high somatostatin-receptor expression for selection of candidates for targeted peptide receptor radionuclide therapy (PRRT) using ^77^Lu or ^90^Y [11]. Additionally, various studies have shown that ^68^Ga-DOTA-SSA PET/CT has a high sensitivity for detecting both lymphatic and distant metastases and has a significant prognostic value and a standardized tumor-uptake value associated with tumor differentiation, Ki-67, and survival [12,13,14]. ^68^Ga-DOTA-SSA PET/CT strongly influences selection of the optimal treatment strategy, leading to management changes in 50–60% of cases [15,16]. ^18^F-Fluorodeoxyglucose positron emission tomography (^18^F-FDG PET/CT) has a low sensitivity for well-differentiated tumors and may help assess high-grade tumors that often display low somatostatin-receptor expression [11]. However, as some high-grade tumors still benefit from ^68^Ga-DOTA-SSA PET/CT, further research is needed to select patients that should be assessed using both ^18^F-FDG PET/CT and ^68^Ga-DOTA-SSA PET/CT.

A laboratory workup uses specific hormones secreted by functional PanNETs and biomarkers common to both functional and nonfunctional PanNETs. Chromogranin A (CgA) is found in the secretory granules of all neuroendocrine cells and is one of the most widely used biomarkers for the diagnosis of neuroendocrine tumors. A meta-analysis conducted by Yang et al. reported a sensitivity of 73% and a specificity of 95% for CgA in the diagnosis of neuroendocrine tumors [17]. Additionally, chromogranin A may be used to monitor therapeutic response, and studies have found CgA levels to be correlated with disease burden and survival [18]. Neuron-specific enolase (NSE) has limited sensitivity and is usually used in conjunction with CgA. CgA/NSE levels were shown to be a prognostic factor for treatment outcome and survival [19,20]. Pancreatic polypeptide is a neuroendocrine differentiation marker with limited sensitivity and specificity. However, when associated with CgA or in the case of metastatic disease, it has increased sensitivity and may have a role in predicting disease control [19,21]. The evaluation of specific biomarkers is necessary for patients presenting with clinical symptoms suggestive of functional PanNETs. Blood levels of bioactive peptides, such as insulin, glucagon, gastrin, somatostatin, vasoactive intestinal peptide (VIP), or other ectopic hormones (growth hormone, adrenocorticotropic hormone (ACTH), are essential for diagnosis and provide prognostic information [21].

### 2.2. Treatment

#### 2.2.1. Surgery

Pancreatic neuroendocrine tumor treatment frequently requires a personalized and multidisciplinary approach (not limited to surgery) that involves somatostatin analogs, targeted radionuclide therapy, and chemotherapy. For most patients, surgical resection is the only potentially curative treatment, especially for localized diseases. However, it also provides a significant benefit in terms of survival and quality of life in patients with metastatic functional tumors. Therefore, it should be considered the treatment of choice whenever possible [22,23]. Surgery is influenced by the tumor’s location, size, functional status, tumor grade, and the presence of lymphatic or distant metastases. Due to their indolent growth, observation may be an option for small, nonfunctional PanNETs < 2 cm. The surgical complication rate must be balanced with the risk of tumor growth and disease progression for these patients. Nevertheless, future long-term studies are necessary to further validate this approach [24,25,26]. Nonfunctional tumors >2cm and all functional tumors should be resected whenever possible, regardless of the size. Enucleation may be considered for small and well-defined tumors without evidence of lymphatic or distant metastases that are located distally to the pancreatic duct. Although this technique is linked to a lower incidence of pancreatic insufficiency, it leads to a higher rate of postoperative pancreatic fistulas. Moreover, these procedures are usually not followed by lymphadenectomy, which portrays useful prognostic information [24,27,28]. Pancreatic resection is the treatment of choice for the remaining patients with resectable disease and usually consists of pancreatoduodenectomy, as well as distal or central pancreatectomy. Lymphadenectomy is recommended, since 24% of patients with grade 1 tumors > 1 cm have lymphatic metastases, and lymphatic involvement carries important prognostic value [29,30,31]. Surgery plays an essential part in the treatment of patients with metastatic disease. Complete resection of hepatic metastases has been linked to improved symptom control and better survival, with five-year survival rates of 50–70% for patients who underwent resection, compared to 30% for those without resection [32,33]. Hepatic debulking surgery can be considered for patients with functional tumors and uncontrolled symptoms, nonfunctional tumors with stable disease, as well as symptoms due to disease burden [34]. Primary tumor resection can be considered for selected patients with unresectable liver metastases, as it has been associated with improved survival [35]. Local ablative therapies can also be used in patients with unresectable metastatic disease, alone or in combination with liver resection [36].

#### 2.2.2. Somatostatin Analogs

For advanced unresectable and metastatic cases, treatment is focused on two main objectives, including symptom control and tackling disease progression. For patients with functional tumors, controlling hormone overexpression and the clinical syndromes derived from hormonal excess are the main therapeutic targets associated with significant morbidity, mortality, and decreased quality of life. Somatostatin analogs (SSA) are usually the first-line therapy for symptom control in functional PanNETs. SSAs bind somatostatin receptors with high affinity and improve clinical symptoms. Moreover, they effectively control diarrhea and VIP blood levels in VIPoma, ensure fast relief control of diarrhea and migratory necrolytic erythema for patients with glucagonomas, inhibit gastrin secretion, and, alongside proton pump inhibitors, ensure symptom control for gastrinomas. They also provide efficient symptom palliation in somatostatinoma [37]. However, due to varying expression of type 2 somatostatin receptors, patients with insulinomas often have a poor response to SSAs, ensuring hypoglycemia control in only 57% of the patients [38]. In insulinomas without somatostatin-receptor expression, SSAs can lead to more hypoglycemic episodes by altering counter-regulatory mechanisms. Additional treatment options consist of high-dose proton pump inhibitors and H2 blockers for gastrinomas, as well as insulin suppression with diazoxide for insulinomas. Telotristat ethyl, an oral tryptophan inhibitor, showed significant and durable control of diarrhea in the TELESTAR trial and is approved for the treatment of refractory carcinoid-syndrome-associated diarrhea in patients with SSA treatment [39]. Peptide receptor radionuclide therapy can also be considered in patients with uncontrolled symptoms, and everolimus may be an option for metastatic insulinoma patients [40].

Somatostatin analogs are efficient in controlling symptoms resulting from hormonal overproduction; however, their clinical effectiveness is not limited to symptom control. Two clinical trials, PROMID [41] and CLARINET [42], demonstrated SSA efficiency in controlling tumor growth; in addition, both lanreotide and octreotide showed superior progression-free survival (PFS) compared to placebo and are considered the first-line treatment for slow-growing (Ki-67 < 10%) PanNETs with somatostatin-receptor expression [34,40]. SSA treatment is generally well-tolerated, with side effects including nausea, bloating, steatorrhea, and cholelithiasis. Currently, timing of SSA treatment initiation remains a matter of debate for stable, nonfunctional, low-burden, and well-differentiated G1 tumors, for which, given their indolent evolution, an initial watchful waiting strategy can be considered [40,43]. Ki-67 < 5%, disease stability prior to treatment initiation, low-hepatic disease burden <25%, and absence of symptoms have been validated as predictive factors for SSA treatment response [44,45].

#### 2.2.3. Other Systemic Therapies

Currently, two targeted therapies are approved for the second-line treatment of PanNETs. The mammalian target of rapamycin (mTOR) inhibitor everolimus demonstrated superiority in terms of PFS and a trend towards an overall survival (OS) advantage, compared to placebo, in patients with progressive low- and intermediate-grade PanNETs in a prospective randomized phase III clinical trial, RADIANT-3 [46]. Sunitinib, a multitargeted tyrosine kinase inhibitor (TKI), also displayed superiority in terms of PFS, compared to placebo, in a randomized phase III trial [47], leading to its approval for the treatment of advanced, progressive, well-differentiated PanNETs. Although sunitinib is the only currently approved TKI, PanNETs are highly vascular tumors, and other TKIs targeting angiogenesis, such as pazopanib, lenvatinib, axitinib, and surufatinib have shown promising activity in phase I/II trials; however, there is a need for further validation in larger trials [48].

Cytotoxic chemotherapy can be considered in progressive, bulky, and well-differentiated PanNETs, as well as high-grade tumors. Currently, there is no consensus on the best chemotherapy regimen, and possible agents include 5-FU, capecitabine, dacarbazine, streptozocin, temozolomide, and oxaliplatin [40,43]. Streptozocin and 5-FU combinations have shown good and durable responses, with objective response rates of 28–42% that can be considered upfront in patients with a high tumor burden [49,50]. Temozolomide has also been efficient in PanNETs, with response rates similar to those of streptozocin, and can be used as an alternative alone or combined with capecitabine [51]. Data from a prospective phase II trial comparing temozolomide alone or in combination with capecitabine showed superior PFS (14.4 vs. 22.7 months) and objective response rates (27.8% vs. 33.3%). Moreover, preliminary overall survival analysis favored the combination; nonetheless, the median OS has not been reached so far in the combined treatment group [52].

#### 2.2.4. Peptide Receptor Radionuclide Therapy

Peptide receptor radionuclide therapy consists of the systemic delivery of radionuclides that bind to cells with high somatostatin-receptor expression, providing targeted radiation to the tumor, leading to cytotoxicity. The most frequently used radioisotopes are β-emitting yttrium (^90^Y) and ^177^lutetium (^177^Lu), both superior in terms of objective response rate, compared to the previous generation of γ-emitting ^111^indium [53]. The main toxicities of both ^90^Y and ^177^Lu are myelotoxicity and nephrotoxicity, with a much more favorable renal toxicity profile for ^177^Lu. A multicentric phase III trial, NETTER-1, compared ^177^Lu-DOTATATE in association with octreotide versus octreotide alone in 229 patients with previously treated well-differentiated midgut neuroendocrine tumors and showed superior PFS, response rate, symptom control, and quality of life [54,55]. The NETTER-1 trial did not include patients with PanNETs; however, several other studies examined PRRT therapy in PanNETs, with median PFS varying from 20 to 39 months and OS ranging from 37 to 79 months [56]. Additionally, the ^177^Lu-DOTATATE treatment also provided symptom control and increased quality of life in patients with PanNETs [57]. Current guidelines recommend PRRT for patients with high somatostatin-receptor expression after the failure of SSA, targeted therapy, and chemotherapy. Furthermore, for patients with functional tumors, SSA should be associated with PRRT to prevent symptom exacerbation or hormonal crisis after PRRT [34,40,43]. 

The increasing availability of accurate diagnostic procedures has led to higher detection rates and earlier diagnosis of PanNETs. However, in recent years, there has been a slight improvement in the prognostic of these patients, particularly for those with advanced disease, where most therapeutic options often achieve only symptom palliation and disease stabilization.

Considering this, the current review points out the main genetic and miRNA-based epigenetic alterations in PanNETs and how these approaches could improve the management of PanNET patients.

## 3. Microbiome and Microbiota in PanNETs

The commensal bacteria, viruses, and fungi that colonize the epithelial surfaces of the human body comprise the microbiota. The gastrointestinal tract is the most significant reservoir in the human body, hosting more than 10^14^ microorganisms encoding more than 5,000,000 genes. This genetic material forms the gut microbiome. The gut microbiota play a crucial role in maintaining the integrity of the intestinal mucosa, protecting against invasive pathogens, and providing essential nutrients, such as vitamins. Furthermore, through a continuous interplay with the enteral epithelial and immune cells, the microbiome assures the proper development of the immune system [58,59]. Disruption of microbiota homeostasis has been linked to the development of several diseases, such as inflammatory bowel disease, irritable bowel syndrome, fatty liver disease, diabetes, obesity, and cardiovascular, autoimmune, neurologic, and psychiatric disorders [60]. Dysbiosis has also been associated with several cancers. Clinical and preclinical data reveal the involvement of microbiota and associated metabolites in all stages of carcinogenesis. Adherent-invasive *Escherichia coli, Helicobacter pylori,* and *Helicobacter hepaticus* are responsible for DNA damage, genomic instability, and impaired DNA repair, thus favoring cancer initiation. *Schistosoma haematobium Clonorchis sinensis* and *Enterotoxigenic Bacteroides fragilis* are involved in cancer promotion through cellular proliferation, antiapoptotic signals, and tumor-promoting inflammation. At the same time, *Fusobacterium nucleatum* supports progression via immune evasion and proliferative stimuli [61,62].

Alterations of the microbiota have also been associated with pancreatic disorders, such as acute and chronic pancreatitis [63,64], type 1 diabetes [65], preneoplastic pancreatic lesions [66,67], and pancreatic cancer. Microbiota are implicated in pancreatic cancer oncogenesis by suppressing the innate and adaptive immune systems and through upregulation of carcinogenetic cellular pathways [68]. Murine models were used to analyze microbiome involvement in pancreatic adenocarcinoma carcinogenesis. Germ-free mice and mice treated with ablative oral antibiotics displayed slower tumor progression than control cohorts. Additionally, fecal transplantation from pancreatic-cancer controls led to accelerated disease progression. Microbial ablation was associated with myeloid-derived suppressor-cell reduction, increased M1 tumor-associated macrophage expression, TH1 differentiation of CD4+ T cells, and CD8+ T-cell activation. Furthermore, bacterial ablation increased PD-1 expression on effector T cells [69]. The binding of micro-organism-associated molecular patterns to specific Toll-like receptors leads to the activation of NF-κB and MAPK pathways, key promoters of pancreatic-cancer oncogenesis [70,71,72].

Few data are available concerning the involvement of microbiota in pancreatic neuroendocrine tumors. *Helicobacter pylori* infection appears to be involved in the development of gastric neuroendocrine tumors, and altered *Faecalibacterium prausnitzii* was observed in patients with midgut neuroendocrine tumors [73]. Furthermore, alterations of gut microbiota modulate tryptophan levels, the precursor of serotonin, a critical neuroendocrine cell effector responsible for the carcinoid syndrome associated with functional tumors [74]. Microbiota can also influence chemotherapy response, which may represent a pathway to increase the efficiency of checkpoint inhibitors [68,75].

## 4. Circulating Tumor Cells in PanNETs

Pancreatic neuroendocrine tumors are a heterogeneous type of cancer concerning clinical behavior, treatment response, and prognosis. Several biomarkers, such as tumor grade, Ki-67 index, and CgA, aid in the clinical management of PanNET patients; however, there is a need to identify novel, reliable, and accessible biomarkers.

Circulating tumor cells (CTCs) are cancer cells that are shed into the blood from the primary tumor site and metastatic lesion. The biogenesis of CTCs is a three-step process: intravasation, migration, and extravasation. Tumor cells undergo an epithelial-mesenchymal transformation, enter the bloodstream, and travel to appropriate distal sites, where they form novel metastatic niches. Circulating tumor cells may travel independently or form clusters/microemboli [76]. 

CTCs are promising biomarkers in metastatic and nonmetastatic cancers, such as breast, colon, prostate, lung, or pancreatic cancer. Several studies have validated their utility in establishing prognosis, monitoring treatment response, and guiding treatment selection [77]. In addition, based on their prognostic value in other human malignancies, CTCs have also been investigated in neuroendocrine cancers, including PanNETs. Khan et al. were the first to analyze whether CTCs can be detected in metastatic neuroendocrine cancers, whether they express epithelial cell-adhesion molecules, and whether they predict radiological progression. Using the Cell Search platform, the authors analyzed blood samples from 79 patients with neuroendocrine tumors, including 19 patients with PanNETs. CTCs were detected in 21% of PanNET samples. CTC levels correlated with urinary 5-hydroxy indole acetic acid (5-HIAA) and liver metastases burden. There was a strong association between stable disease and a lack of CTCs [78]. The authors performed a second study using samples from 175 patients with neuroendocrine neoplasms; 42 patients had PanNETs. The presence of CTCs was linked to increased tumor burden, higher tumor grade, and elevated seric CgA. CTCs were also associated with worse PFS and OS, and they were able to define poor prognostic subgroups within the tumor-grade groups [79]. To evaluate the ability of CTCs to predict treatment response, the same authors analyzed before- and after-treatment samples from 138 metastatic neuroendocrine neoplasm-31 PanNETs. Patients without CTCs and those with a >50% reduction in CTC count after treatment had better outcomes. CTC variations were strongly correlated with OS (HR 4.13, *p* = 0.0002) [80,81].

The presence of CTCs was also correlated with skeletal metastases. Rizzo et al. investigated samples from 254 patients, including 119 patients with PanNETs. The authors demonstrated a significant association between the presence of CTCs and bone metastases (*p* < 0.0001). A cutoff value of ≥2 circulating tumor cells was able to identify patients with skeletal involvement in PanNETs. CTCs were also associated with tumor grade but not lung, lymphatic, or peritoneal metastases [82].

## 5. Genetic Insights in PanNETs

Pancreatic neuroendocrine tumors represent a heterogeneous group of tumors that most often arise sporadically. However, 10% of PanNETs may occur in hereditary syndromes, such as multiple endocrine neoplasia type 1 (MEN1), Von Hippel—Lindau disease (VHL), type 1 neurofibromatosis (NF1), and tuberous sclerosis complex (TSC). MEN1 is the most frequently occuring hereditary syndrome associated with PanNETs. Clinically, the disorder is characterized by two or more gastroenteropancreatic tumors, parathyroid adenoma, or pituitary adenoma. PanNETs in MEN1 patients are often nonfunctional and can be microscopically identified in 80–100% of cases. Gastrinomas are the most frequent functional tumors, followed by insulinomas [83]. Patients with MEN1 present with germline mutations in the MEN1 tumor-suppressor gene, which encodes menin and is located on the 11q13 chromosome, resulting in a menin functional deficiency. More than 1336 specific mutations of the MEN1 gene have been described. However, consistent with Kudson’s 2-hit hypothesis, germline mutations are not the only mutations involved in MEN1 pathogenesis. Furthermore, somatic mutations, including loss of heterozygosity (LOH), point mutations, and intragenic mutations, have been identified in PanNET patients [84]. MEN1 mutations have also been described in sporadic PanNETs. Scarpa et al. underlined the presence of MEN1 mutations in 37% of sporadic PanNET patients [85]. MEN1 has an important role in PanNET pathogenesis, as it is involved in the homeostasis of several cellular-proliferation pathways. In addition, it has multiple roles, including the inhibition of wingless integration 1 (Wnt1)/β-catenin [86] and Hedgehog [87] pathway signaling, modulation of nuclear-factor kappa B transactivation (NF-kB) [88], or inhibition of proliferation by blocking mitogen-activated protein kinase and extracellular signal-regulated kinase (MAPK-ERK) [89]. MEN1 could also act as a phosphatidylinositol-3-kinase (PI3K)- Akt- mTOR inhibitor by Akt 1 downregulation [90].

PanNETs also occur in VHL disease, alongside pheochromocytoma, renal cell carcinoma, and cerebellar or retinal hemangioblastomas. VHL disease is an autosomal dominant disorder caused by mutations in the VHL gene located on chromosome 3p25. Mutations of the tumor suppressor VHL gene lead to overexpression of hypoxia-inducible factors and enhanced hypoxia-driven angiogenesis. VHL mutations are rare in sporadic PanNETs; however, a study by Schmitt et al. linked VHL mutation via deletion or promoter methylation to increased hypoxia signaling and adverse outcomes [83,91].

Neurofibromatosis type 1 patients have an increased risk of developing various types of cancers, such as myeloid leukemia, central nervous system tumors, pheochromocytoma, rhabdomyosarcomas, and PanNETs. Gastroenteropancreatic neuroendocrine tumors can be detected in about 10% of NF1 patients [92]. Mutations in the NF1 tumor suppressor gene situated on the 17q11.2 chromosome, as well as the loss of NF1 function, promote proliferation by increasing activity in the RAS-MAPK and PI3K-Akt-mTOR pathways [93]. Tuberous sclerosis complex, an autosomal dominant disease, can also be associated with an increased risk of PanNETs. Mutations in the TSC1 and TSC2 genes lead to amplified proliferation via mTOR pathway activation [94]. Scarpa et al. [85] and Jiao et al. [95] identified mutations in the TSC 1 and 2 genes as a mechanism for mTOR pathway dysregulation in sporadic PanNETs.

The MEN1 gene is the most frequently mutated gene in sporadic PanNETs. Studies reported MEN1 mutations in 37–44% of sporadic PanNETs, underlying its major role in PanNET oncogenesis [85,95]. Furthermore, Jiao et al. [95] found two other mutually exclusive genes that were mutated in over 40% of sporadic PanNETs, namely the death-domain-associated protein (DAXX) and α thalassemia/mental retardation syndrome X-linked (ATRX) genes. ATRX and DAXX are involved in chromatin remodeling and apoptosis. Mutations in the ATRX and DAXX genes and the subsequent loss of function lead to chromosomal instability and the alternative lengthening of telomeres (ALT). This process leads to telomeric elongation in a telomerase-independent way through a homologous recombination mechanism [96,97]. ATRX/DAXX mutation and subsequent ALT is a late event in tumor development [97]. The presence of MEN1, DAXX, and ATRX mutations or combinations of MEN1 and ATRX/DAXX mutations was associated with better outcomes, compared to PanNETs lacking these mutations [95].

About 25% of sporadic PanNET patients present with dysregulation of the mTOR signaling pathway due to mutations in PTEN, TSC1, TSC2, PIK3CA, and DEPDC5 genes [85,95]. Missiaglia et al. found low TSC2 and PTEN expression to be associated with tumor functional status, increased tumor aggressiveness, and worse survival [98]. Additionally, whole-genome sequencing studies found mutations in the base-excision repair Mut Y homolog (MUTYH) gene and homologous repair breast cancer 2 (BRCA2) and checkpoint kinase 2 (CHEK2) genes [85]. In a significant number of cases, the presence of MUTYH/CHEK2 in association with DAXX/ATRX but not MEN1 mutations implies a possible MEN-1-independent carcinogenetic mechanism in those PanNETs [92].

Alterations of the epigenetic mechanisms, including DNA methylation and microRNA (miRNA) expression, have been associated with PanNET pathogenesis. DNA methylation, via promoter hypermethylation, could lead to the suppression of the tumor-suppressor gene. Several studies have reported altered DNA methylation in PanNETs. Ras association domain family 1 (RASSF1) is a tumor-suppressor gene frequently mutated in various human cancers. Loss of function is associated with cell-cycle dysregulation, increased genetic instability, cellular mobility, and resistance to apoptotic signals. RASSF1 mutations, through promoter hypermethylation, have been described in 75% of PanNETs [99,100]. Cyclin-dependent kinase inhibitor 2A (CDKN2A) encodes the tumor-suppressor protein p16, involved in cell-cycle regulation. CDKN2A mutations have been reported in 40% of PanNETs and were associated with a poor prognosis [100]. Mutations in CDKN2A may be expressed in up to 52% of gastrinomas, while they are rare events in insulinomas [101,102]. A tumor-suppressor tissue inhibitor of metalloproteinase-3 (TIMP3) methylation was identified by Wild et al. in 44% of PanNETs. Patients with hepatic or lymphatic metastases were more likely to have TIMP3 mutations (79% vs. 14%) [103]. The CpG island methylator phenotype (CIMP), a gene-silencing mechanism often present in tumors harboring promoter methylation, was found in most PanNETs and was associated with poor outcomes [92,104]. 

## 6. MiRNA-Based Epigenetic Challenges in PanNETs

Among epigenetic mechanisms, miRNA dysregulation has been incriminated in the carcinogenesis of various malignancies. MiRNAs are small, noncoding RNA (18-25 nucleotides) that modulate gene expression post-transcriptionally by biding messenger RNA (mRNA), with subsequent translation inhibition and downregulation of gene expression, by complementary base pairing, leading to post-transcriptional gene silencing [7,105,106] (Figure 1).

With over 2500 mature human miRNAs registered, they control approximately 30% of all protein-encoding gene expression [107,108]. In addition to intracellular miRNA, various cells may actively secrete miRNA through exosomes and conjugate to RNA binding proteins, or they may exhibit passive secretion of miRNA after apoptosis or tissue injury [109]. MiRNAs can be detected in various biological fluids, such as blood, saliva, urine, bronchial lavage, peritoneal and pleural fluids, cerebrospinal fluid, and other body-fluid types [110]. MiRNA can be quantified through quantitative real-time PCR, microarray, and next-generation sequencing, as well as in situ hybridization, and have the potential to become a new class of minimally invasive biomarkers. Moreover, they can also become potential therapeutic targets [109,111].

Despite extensive research on the involvement of miRNAs in the diagnosis, prognosis, and treatment of various cancers, few data concerning their role in pancreatic neuroendocrine tumors are available. We will further critically appraise the existing evidence concerning the involvement of miRNAs as biomarkers in PanNETs.

MiRNAs biogenesis is starts in the nucleus with long hairpin transcripts named pri-miRNAs, which are then enzymatically cleaved to smaller transcripts containing about 70nucleotides, called precursor miRNAs(pre-miRNAs). After their export in the cytoplasm, pre-miRNAs are processed in single-strand mature miRNA. Then, by incorporation into the RNA-induced silencing complex (RISC), mature miRNA is functionalized and guided to silence specific mRNA transcripts by degradation or translational repression.

### 6.1. Brief Overview of miRNA in PanNETs

miRNA-21 upregulation has been observed in various cancers, including ovarian, lung, and hepatocellular carcinomas, as well as head and neck cancers [112]. STAT3 and NFκB transcription factors are frequently involved in the pathogenesis of various human cancers, influencing critical processes, such as proliferation, invasion, and apoptosis. Both NFκB and STAT3 modulate mir-21 levels. Research involving cellular lines and animal models identified the role of miRNA-21 as an oncogenesis promoter by targeting the expression of PTEN, programmed cell death 4 (PDCD4), insulin growth factor binding protein 3, and F-box only protein 11 [113]. Several studies have underlined its involvement in PanNETs. Grolmusz et al. analyzed the miRNA expression profiles of 40 PanNET patients and found higher expression of miR-21, miR-10a, and miR-106 in grades 2 and 3, compared to grade 1 tumors. Furthermore, multivariate analysis found miR-21 to be an independent prognostic factor, with higher levels linked to worse PFS and OS [114]. miR-21 overexpression was also associated with metastatic disease, particularly with liver metastases and higher Ki67 levels [115,116]. 

The miR-30 family comprises six members, including miR-30a, miR-30b, miR-30c-1, miR-30c-2, miR-30d, and miR-30e. Dysregulation of the miR-30 family has been associated with numerous cancers, such as lung, breast, gastric, pancreatic, colorectal, ovarian, and cervical cancers [117]. Research has shown that the miR-30 family may act as an oncogene or tumor-suppressor gene, depending on the primary tumor location. MiR-30a is a potent inhibitor of cancer-cell proliferation by targeting essential pathways, such as the Wnt/β-catenin, PTEN/Akt, and PI3K pathways. MiR-30a also downregulates expression of BCL-2 and Beclin 1 and promotes apoptosis [118]. However, miR-30a may also act as a tumorigenesis promoter in ovarian cancer or melanoma [117]. Similarly, Kim et al. demonstrated the role of miR-30a in promoting cancer-cell growth in PanNETs by competing with human antigen D and downregulating p27 expression, inhibiting cyclin-dependent kinase activity and cell proliferation [119]. MiR-30a-5p expression was also linked to metastatic disease in PanNETs [115].

Several studies have reported the role of miR-210 as either a tumor suppressor by inhibiting proliferation and tumor progression or as an oncogene. Moreover, it plays an essential role in carcinogenesis, and it is a key regulator of hypoxia by promoting hypoxia-inducible factor 1α activity through a mechanism involving a positive feedback loop [120]. In PanNETs, miR-210 has prognostic significance, as its expression has been linked to metastatic disease [115,121].

MiR-183 has been studied in various cancer types, and current data highlight its function in essential cancer-related processes, such as proliferation, invasion, migration, angiogenesis, epithelial-mesenchymal transition (EMT), and apoptosis [122]. One study confirmed the oncogenic properties of miR-183 on pancreatic neuroendocrine cell lines. miR-183 abolished the tumor suppressor function of MEG3 and upregulated the tumor-promoting function of the BRI3 gene [123]. Michael et al. underlined the dynamic nature of the involvement of miRNAs in PanNET tumorigenesis. Using a genetically engineered mouse model, they analyzed a set of miRNAs previously found to be upregulated in metastatic PanNETs. MiR-137 was found to enable tumor growth and invasion; moreover, miR-23Bb promoted metastasis, whereas a cluster formed by miR-130/301 favored evasion of proapoptotic signaling [124].

### 6.2. Clinical Implications of miRNAs in PanNETs

#### 6.2.1. The Prognostic Role of miRNAs in PanNETs

Few data are available regarding miRNAs as biomarkers for PanNETs. Several studies have investigated miRNAs as potential prognostic, diagnostic, or therapeutic biomarkers (Table 1). Zhou et al. found miR-183-5p expression to be upregulated in PanNETs [125]. An analysis of 57 PanNETs correlated miR-183-5p with increased tumor size, grade, and serotonin levels, as well as lower somatostatin receptor 2 expression and superior survival. The same study also analyzed miRNA132-3p, miRNA145-5p, miRNA34a-5p, and miRNA449a; in addition, it linked miR-132-3p to increased somatostatin expression and low-grade tumors, as well as vascular invasion. miR-34a-5p was linked to hormone expression. miRs 145-5p and miR-449a were associated with a poor prognosis [126]. Cavalcanti et al. [127] demonstrated that high levels of miR-96-5p and miR-130b-3p expression were linked to higher tumor grade. In contrast, higher miR-194-5p levels were associated with lower tumor grades, underlying a potential role for miRNAs in aiding histopathological analysis and clinical decisions for PanNET patients. miR-196a was also proven to be of prognostic value for patients with PanNETs. The expression of eight miRNAs was evaluated in 37 resected PanNETs, and miR-196a was significantly associated with higher T, N stage, Ki67, and mitotic count, as well as worse survival [128]. Zimmerman et al. [115] analyzed tissue samples from 37 PanNETs and found metastatic disease to be associated with miR-30a-5p, miR-210, miR-339-3p, miR-345, and miR-660 overexpression; in addition, miR-150, miR-21, and miR-660 levels correlated with proliferation index. MiR-3653 portrays a poor prognosis, as its upregulation is associated with metastatic PanNETs [129].

#### 6.2.2. The Diagnostic Role of miRNAs in PanNETs

MiRNAs may also aid in the diagnosis of PanNETs (Table 2). Thorns et al. analyzed the expression profiles of 754 miRNAs and found a signature of 13 miRNAs that distinguished healthy volunteers from PanNETs; however, only miR-193b expression varied significantly. Furthermore, some miRNAs offered additional prognostic information suggesting that miR-642 was correlated with Ki67, while miR-210 was linked to the presence of metastatic disease [121]. Similar data were obtained by Roldo et al., who demonstrated that miR-103 and miR-107 expression, as well as lack of miR-155 expression, differentiates tumors from normal tissue. Furthermore, they also identified a 10-miRNA signature that distinguished endocrine from acinar tumors [116]. In a similar manner, a recent study performed exosomal miRNA profiling on 140 plasma and tissue samples of patients with pancreatic ductal adenocarcinoma, PanNETs, intraductal papillary mucinous neoplasms, and ampulla of Vater carcinomas. According to the results, lesion-specific miRNAs signatures were able to differentiate between pancreatic lesions [130]. MiR-30a-3p expression, alongside miR-24, miR-18a, miR-92a, miR-342-3p, miR-99b, miR-106b, miR-142-3p, and miR-532-3p, enabled the distinction between high-risk PanNETs and other pancreatic cystic lesions requiring resection, as well as other low-risk pancreatic cysts amendable to observation, while an miRNA signature consisting of let-7b-5p, let-7i-5p, miR-143-3p, miR-30d-5p, miR-451a, and miR-486-p increased the diagnostic accuracy of chromogranin A [131,132]. Panarelli et al. [133] confirmed that miRNA expression could aid in the differential diagnosis of neuroendocrine tumors. MiR-429, miR-487b, miR-615, and miR-92b expression have grouped PanNETs as appendiceal, ileal, or rectal neuroendocrine tumors.

#### 6.2.3. The Therapeutic Role of miRNAs in PanNETs

MiRNAs control the expression of thousands of different genes, coordinating multiple cellular pathways. With the progress in cancer profiling, treatment can now be customized for each individual. The complexity of microRNA biology may also provide novel therapeutic options for the management of PanNETs. There are currently two available therapeutic approaches involving miRNAs (Table 3). The first aims to inhibit oncogenic miRNAs using anti-miR locked nucleic acids that bind to their target miRNA and form highly stable complexes that prevent miRNAs from interacting with their target genes. The second strategy involves tumor-suppressor miRNA substitution using miRNA mimics to restore function [111]. Dettori et al. used mouse models to investigate the therapeutic potential of an anti-miR-214 compound in PanNETs, melanoma, and triple-negative breast cancer. Tumors treated with the anti-miR-214 displayed lower levels of circulating tumor cells, as well as fewer lung and lymphatic metastases [134]. MiR-431 promotes tumorigenesis by targeting Ras/Erk signaling and was found to be upregulated in PanNETs. Inhibition of miR-431 by locked nucleic acids leads to significant downregulation of Erk signaling and the invasiveness of PanNET cells [135].

MiRNAs may also be used as biomarkers for treatment response. Yamaha et al. explored the antitumoral effects of metformin on PanNET cell lines and verified the existence of specific miRNA expression profiles correlated with the metformin antitumor effect [136]. MiR-34a, a downstream effector of p53, displays tumor-suppressive properties by promoting cell-cycle arrest and apoptosis. Abnormal miR-34a expression was observed in PanNETs and could also act as a potential therapeutic target [126,131]. Several studies reduced tumor proliferation and induced apoptosis using miR-34a mimics in colon, pancreas, gastric, and breast cancer models, and similar results may be achieved in PanNETs [137].

Bai et al. [135] studied the function of miR-224 in PanNETs and demonstrated its involvement via the proprotein convertase subtilisin/kexin 9 (PCSK9)/glucocorticoid axis in tumorigenesis. Furthermore, the authors underlined potential therapeutic implications for miR-224 by transfecting BON-1 cell lines with miR-224 agomir, which resulted in significant proliferation, invasion suppression, and increased apoptosis. MiRNAs potential as diagnostic, prognostic and therapeutic biomarkers in PanNETs is summarized in Figure 2.

## 7. Conclusions

Pancreatic neuroendocrine tumors are a significant cause of morbidity and mortality, especially in the elderly. Prognosis is influenced by key factors, such as stage, tumor grade, proliferation index, and functional status. Limited-stage disease treatment mainly consists of surgery, while for advanced and metastatic disease, somatostatin analogs, mTOR inhibitors, PRRT, and cytotoxic chemotherapy control symptoms and provide a limited survival benefit. Although most pancreatic neuroendocrine tumors arise sporadically, several genetic mechanisms have been incriminated in tumor oncogenesis. MEN1 mutations can be found in almost half of PanNET tumors, underlying its substantial involvement in tumor development. However, genetic dysregulation in PanNETs is not limited to MEN1 mutations; moreover, 40% of tumors are ATRX/DAXX-mutated, and mTOR mutations are present in 25% of cases. Furthermore, MUTYH/CHEK2, BRCA2, CDKN2A, and TIMP3 have also been associated with pancreatic neuroendocrine tumors. Nonetheless, genetic mechanisms are not the only mechanisms incriminated in tumorigenesis. Epigenetic mechanisms, such as genetic silencing via DNA hypermethylation and miRNA dysregulation, have also been linked to the pathogenesis of PanNETs.

MiRNAs modulate the expression of more than one-third of all genes, and they are involved in a plethora of processes that assure homeostasis. MiRNA dysregulation is an important part of carcinogenesis, and in recent years, as new research has emerged, their function as oncogenes or tumor suppressors has become more and more evident. Despite the small number of studies and heterogeneous data regarding the involvement of MiRNAs in pancreatic neuroendocrine tumors, they are implicated in PanNET oncogenesis and are promising diagnostic, prognostic, and therapeutic biomarkers. Several miRNAs provide essential prognostic information that may result in better patient risk profiles. Furthermore, miRNA signatures were able to aid in the differential diagnosis of various pancreatic lesions and distinguish tumor samples from healthy controls. MiRNAs could also change the management of PanNETs as biomarkers that may help in treatment selection or response evaluation; however, they are also potential therapeutic targets. On the other hand, despite their potential, more research in the field of bioinformatics, miRNA profiling, and in vitro and in vivo preclinical research models is needed to better understand the various pathways of miRNA dysregulation. Furthermore, there is a demand for better funding and larger trials to validate miRNAs as biomarkers or therapeutic targets in PanNETs.

## Figures and Tables

**Figure 1 ijms-23-01109-f001:**
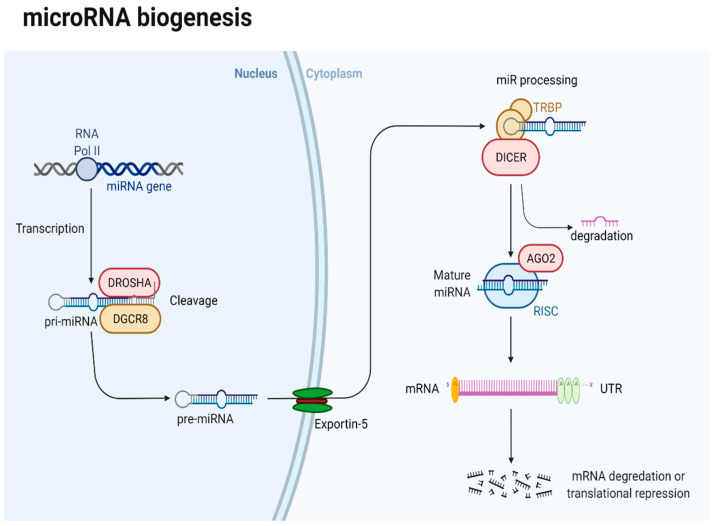
Schematic illustration of microRNA biogenesis.

**Figure 2 ijms-23-01109-f002:**
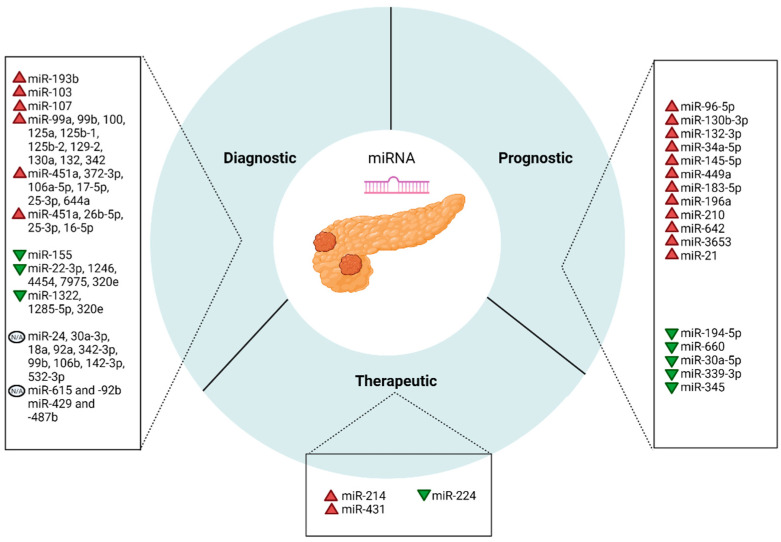
Schematic illustration of the effect of miRNAs as diagnostic, prognostic, and therapeutic biomarkers in PanNETs.

**Table 1 ijms-23-01109-t001:** The prognostic role of miRNAs in PanNETs.

miRNA	Expression	Function	Clinical Implication	Ref. No.
miR-96-5p	⬆	Oncogenic, FoxO1a inhibition	High tumor grade	[127]
miR-130b-3p	⬆	N/A		
miR-194-5p	⬇	N/A		
miR-132-3p	⬆	Tumor-suppressing and tumor-promoting function	Low tumor grade Vascular invasion Somatostatin expression	[126]
miR-34a-5p	⬆	N/A	Somatostatin, gastrin, and serotonin expression	
miR-145-5p	⬆	N/A	High tumor grade Lymphatic invasion Serotonin expression	
miR-449a	⬆	Oncogenic via histone deacetylases 3/4	High tumor grade, mitotic and proliferative activity, lymph-node invasion	
miR-183-5p	⬆	Tumor suppressor	High tumor grade Tumor size Somatostatin-receptor expression	
miR-196a	⬆	N/A	Advanced tumor Lymph-node invasion High mitotic and proliferative activity Recurrence	[128]
miR-660 miR-30a-5p miR-339-3p miR-345	⬇	N/A	Metastatic disease	[115]
miR-210	⬆	Oncogenic	Metastatic disease	[115,121]
miR-642	⬆	Oncogenic	Proliferative activity	[121]
miR-3653	⬆	Oncogenic associated with ATRX mutations	Metastatic disease	[129]
miR-21	⬆	Oncogenic	Tumor grade Metastatic disease Proliferative activity	[115] [114] [116]

Forkhead Box Protein O1A (FoxO1a), phosphatase and tensin homolog (PTEN), alpha thalassemia/mental retardation syndrome x-linked (ATRX), epithelial-mesenchymal transformation (EMT).

**Table 2 ijms-23-01109-t002:** The diagnostic role of miRNAs in PanNETs.

miRNA	Expression	Sample	Sample Size	Diagnostic Implication	Ref. No.
miR-193b	⬆	Serum, FFPE	37	Healthy vs PanNETs	[121]
miR-103 miR-107	⬆	FFPE	96	Healthy vs. tumor	[116]
miR-155	⬇				
miR-99a, 99b, 100, 125a, 125b-1, 125b-2, 129-2, 130a, 132, 342	⬆			PanNETs vs. pancreatic adenocarcinoma/normal pancreas	
miR-451a, 372-3p, 106a-5p, 17-5p, 25-3p, 644a	⬆	Serum, FFPE	140	PanNETs vs. chronic pancreatitis	[130]
miR-22-3p, 1246, 4454, 7975, 320e	⬇				
miR-451a, 26b-5p, 25-3p, 16-5p	⬆			PanNETs vs. pancreatic adenocarcinoma	
miR-1322, 1285-5p, 320e	⬇				
miR-24, 30a-3p, 18a, 92a, 342-3p, 99b, 106b, 142-3p, 532-3p	N/A	FFPE	120	High-grade IPMNs, cystic PanNETs, and SPN vs. low-grade IPMN, SCA	[110]
miR-615 and -92b miR-429 and -487b	N/A	FFPE	81	PanNETs vs. ileal, appendicular, rectal neuroendocrine tumors	[133]

IPNM, intraductal papillary mucinous neoplasms; SPN, solid pseudopapillary neoplasms; SCA, serous cystadenomas; FFPE, formalin-fixed paraffin-embedded tissue.

**Table 3 ijms-23-01109-t003:** The therapeutic role of miRNAs in PanNETs.

miRNA	Expression	Model	Treatment	Outcomes	Ref. No.
miR-214	⬆	Mouse	miR-214 inhibition	-Reduced tumor volume -Decreased volume of peripancreatic lymphatic metastases -Reduced tumor vascularization	[134]
miR-431	⬆	Cell lines	miR-431-targeted locked nucleic acids	-Reduced invasiveness	[135]
miR-224	⬇	Cell lines	miR-224 agomir	-Promotes apoptosis -Inhibits proliferation, invasion	[136]

## Data Availability

Not applicable.

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
