# Peer review of "Current and New Challenges in the Management of Pancreatic Neuroendocrine Tumors: The Role of miRNA-Based Approaches as New Reliable Biomarkers"

_ijms, 2022, doi:10.3390/ijms23031109_

Round 1
Reviewer 1 Report
Andrei Havasi et al. reviewed the challenges in the management of pancreatic neuroendocrine tumors (PanNETs) and the potential roles of microRNAs as markers and therapeutic targets in PanNETs. In the abstract, line 37, it should be “MicroRNAs (miRNAs) are small endogenous ….” instead of “MicroRNAs are small endogenous …..” as the abbreviation miRNAs appears in line 40. In abstract, it has been mentioned PanNET, but as PanNETs in the first line (line 49) of the introduction - it should be PanNETs throughout the manuscript if authors meant for tumors but not for tumor. In line 60, what type of peptides? – needs to elaborate for the readers outside of this research field. Lines 116-118: What kind of data used in meta-analysis? Is it genomic or proteomic or any other data? Line 202: Please double check if there are any other targeted therapies available. In figure 2, three segments (diagnostic, prognostic, and therapeutic) are sufficient; is it necessary to give space for miRNA as fourth segment? It should be in the middle with Pancreas if I am not misunderstood. Line 496: this is figure 2 legend – the sentence “miRNAs are involvement in” requires rephrasing. In conclusions (section 5 lines 526-527) part, the authors mentioned “more research is needed to better understand the various pathways of miRNA dysregulation”. Here, it would be nice to elaborate on what type of research approaches are appropriate in this regard? Proof reading by a Native English speaker is required to improve the quality of the manuscript. Overall, the authors cited recent articles; and the manuscript has been well written and presented in a good way.
Author Response
Manuscript ID:ijms-1539472
Title: “Current and new challenges in the management of pancreatic neuroendocrine tumors. The role of miRNAs-based approaches as new reliable biomarkers” by Havasi et al.
Reviewer 1 comments:
Andrei Havasi et al. reviewed the challenges in the management of pancreatic neuroendocrine tumors (PanNETs) and the potential roles of microRNAs as markers and therapeutic targets in PanNETs.
We want to thank the reviewer for reviewing our manuscript. We are confident that by answering the reviewer’s requests we will improve the current paper. Our main purpose is to present clear data which can be helpful for the clinicians and researchers involved in the field of gastrointestinal tumors.
In the abstract, line 37, it should be “MicroRNAs (miRNAs) are small endogenous ….” instead of “MicroRNAs are small endogenous …..” as the abbreviation miRNAs appears in line 40. In abstract, it has been mentioned PanNET, but as PanNETs in the first line (line 49) of the introduction - it should be PanNETs throughout the manuscript if authors meant for tumors but not for tumor.
We have amended our manuscript according to the reviewer’s suggestions. The changes were done using track changes.
In line 60, what type of peptides? – needs to elaborate for the readers outside of this research field.
Thank you for the comment. We modified the sentence so it can be easy to follow for the readers outside the research field. Now the sentence reads as: “Up to 90% of all PanNETs are nonfunctional or secrete low levels of hormones, not enough to produce symptoms or secrete hormones without clinical manifestations.”
Lines 116-118: What kind of data used in meta-analysis? Is it genomic or proteomic or any other data?
The meta-analysis conducted by Yang et al. used proteomic data in most of the studies. Because of the heterogenic design of the studies included in the meta-analysis we didn’t consider necessary to highlight this aspect.
Line 202: Please double check if there are any other targeted therapies available.
We thank the reviewer for the comment. We double-checked to see if other targeted therapies were granted approval since the date of our data collection for the review and we couldn’t find any new therapies.
In figure 2, three segments (diagnostic, prognostic, and therapeutic) are sufficient; is it necessary to give space for miRNA as fourth segment? It should be in the middle with Pancreas if I am not misunderstood.
We also modified figure 2 according to your suggestions. Now figure 2 is designed as:
Line 496: this is figure 2 legend – the sentence “miRNAs are involvement in” requires rephrasing.
Thank you for the suggestion.We have deleted the sentence so that it doesn’t create confusion for the readers.
In conclusions (section 5 lines 526-527) part, the authors mentioned “more research is needed to better understand the various pathways of miRNA dysregulation”. Here, it would be nice to elaborate on what type of research approaches are appropriate in this regard?
We thank the reviewer for the observation. We have amended our manuscript to include a more ample description of the research needed. Now the sentence can be read as:”On the other hand, despite their big potential, more research in the field of bioinformatics, miRNA profiling, in vitro and in vivo preclinical research models is needed to better understand the various pathways of miRNA dysregulation.”
Proof reading by a Native English speaker is required to improve the quality of the manuscript. Overall, the authors cited recent articles; and the manuscript has been well written and presented in a good way.
The manuscript was sent to our Professional English Editing team for a second round of verification. They also checked for punctuation, spelling and grammar.
Thank you for taking your time and analyzing our manuscript.

Reviewer 2 Report
This is a generally well-written article on pancreatic neuroendocrine tumors and miRNAs. This topic is of great interest. The author can further make this paper better; please see below – just a few points for improvements.
The authors discuss various treatment options but ignore influences of other factors including the microbiome. It has been well known that many environmental and lifestyle factors can influence immune cells, the microbiota, tumor development and response to therapy. The authors should discuss those points. Many factors influence response to therapy in each patient differentially.
Along with these points, research on environment factors, microbiome, immunity, and molecular tissue biomarkers should be pursued. The authors should discuss molecular pathological epidemiology, which can integrate those factors, molecular pathologies, immune response, and clinical outcomes in cancer. Molecular pathological epidemiology is discussed in Annu Rev Pathol 2019, Curr Colorectal Cancer Rep 2017, Hum Genet 2021, etc. Molecular pathological epidemiology research can be a promising direction and improve prediction of response to therapy.
Author Response
Manuscript ID: ijms-1539472
Title: “Current and new challenges in the management of pancreatic neuroendocrine tumors. The role of miRNAs-based approaches as new reliable biomarkers” by Havasi et al.
Reviewer 2 comments:
This is a generally well-written article on pancreatic neuroendocrine tumors and miRNAs. This topic is of great interest. The author can further make this paper better; please see below – just a few points for improvements.
We want to thank the reviewer for the time spent reviewing our manuscript. We are confident that by answering to the reviewer’s requests we will improve the current paper.
The authors discuss various treatment options but ignore influences of other factors including the microbiome. It has been well known that many environmental and lifestyle factors can influence immune cells, the microbiota, tumor development and response to therapy. The authors should discuss those points. Many factors influence response to therapy in each patient differentially.
We appreciate the comment of the reviewer. We have added a new chapter in our manuscript that encompasses the reviewer’s suggestions. The new chapter can be read like:
“3. Microbiome and microbiota in PanNETs
The commensal bacteria, viruses, and fungi that colonize the epithelial surfaces of the human body comprise the microbiota. The gastrointestinal tract is the most significant reservoir in the human body, hosting more than 1014 microorganisms encoding more than 5.000.000 genes. This genetic material forms the gut's microbiome. The gut microbiota plays a crucial role in maintaining the integrity of the intestinal mucosa, protecting against invasive pathogens, and providing essential nutrients such as vitamins. Furthermore, through a continuous interplay with the enteral epithelial and immune cells, the microbiome assures the proper development of the immune system [58,59]. Disruption of the microbiota homeostasis has been linked to the development of several diseases such as inflammatory bowel disease, irritable bowel syndrome, fatty liver disease, diabetes, obesity, cardiovascular, autoimmune, neurologic, and psychiatric disorders [60]. Dysbiosis has also been associated with several cancers. Clinical and preclinical data reveals the microbiota's and its metabolite's involvement in all stages of carcinogenesis. Adherent-invasive Escherichia coli, Helicobacter pylori, and Helicobacter hepaticus are responsible for DNA damage, genomic instability, impaired DNA repair, thus favoring cancer initiation. Schistosoma haematobium Clonorchis sinensis and Enterotoxigenic Bacteroides fragilis are involved in cancer promotion through cellular proliferation, antiapoptotic signals, and tumor-promoting inflammation. At the same time, Fusobacterium nucleatum supports progression via immune evasion and proliferative stimuli [61,62].
Alterations of the microbiota have also been associated with pancreatic disorders such as acute and chronic pancreatitis [63,64], type 1 diabetes [65], preneoplastic pancreatic lesions [66,67], and pancreatic cancer. Microbiota is implicated in pancreatic cancer oncogenesis by suppressing the innate and adaptive immune systems and through upregulation of carcinogenetic cellular pathways [68]. Murine models were used to analyze microbiome involvement in pancreatic adenocarcinoma carcinogenesis. Germ-free mice or mice treated with ablative oral antibiotics displayed slower tumor progression than control cohorts. Additionally, fecal transplantation from pancreatic cancer controls led to accelerated disease progression. Microbial ablation was associated with myeloid-derived suppressor cells reduction, increased M1 tumor-associated macrophage expression, TH1 differentiation of CD4+ T cells, and CD8+ T-cell activation. Furthermore, bacterial ablation increased PD-1 expression on effector T cells [69]. The binding of microorganism-associated molecular patterns to specific Toll-like receptors leads to NF-κB and MAPK pathways activation, key promoters of pancreatic cancer oncogenesis [70–72].
There is little data available on microbiota involvement in pancreatic neuroendocrine tumors. Helicobacter pylori infection appears to be involved in the development of gastric neuroendocrine tumors, and altered Faecalibacterium prausnitzii could be observed in patients with midgut neuroendocrine tumors [73]. Furthermore, gut microbiota alterations modulate tryptophan levels, the precursor of serotonin, a critical neuroendocrine cell effector responsible for the carcinoid syndrome associated with functional tumors [74]. Microbiota can also influence chemotherapy response, and it may represent a pathway to increase checkpoint inhibitors' efficiency [68,75].”
Along with these points, research on environmental factors, microbiome, immunity, and molecular tissue biomarkers should be pursued. The authors should discuss molecular pathological epidemiology, which can integrate those factors, molecular pathologies, immune response, and clinical outcomes in cancer. Molecular pathological epidemiology is discussed in Annu Rev Pathol 2019, Curr Colorectal Cancer Rep 2017, Hum Genet 2021, etc. Molecular pathological epidemiology research can be a promising direction and improve prediction of response to therapy.
We want to thank the reviewer for the suggestion of improving our manuscript. We have amended our manuscript according to your input and added another chapter to include data about circulating tumor cells in PanNETs although literature concerning pancreatic neuroendocrine tumors is scarce. Now the text can be read like:
“4. Circulating tumor cells in PanNETs
Pancreatic neuroendocrine tumors are a heterogeneous type of cancer concerning clinical behavior, treatment response, and prognosis. Several biomarkers such as tumor grade, Ki-67 index, and CgA aid in the clinical management of these patients, however, there is a need to identify novel, reliable, and accessible biomarkers.
Circulating tumor cells (CTCs) are cancer cells that are shed into the blood from the primary tumor site and metastatic lesion. The biogenesis of CTCs is a three stepped process: intravasation, migration, and extravasation. Tumor cells undergo an epithelial-mesenchymal transformation, enter the bloodstream, and travel to appropriate distal sites where they form novel metastatic niches. Circulating tumor cells may travel as a single cell or form clusters – microemboli [76].
CTCs are promising biomarkers in metastatic and non-metastatic cancers such as breast, colon, prostate, lung, or pancreatic cancer. Several studies have validated their utility in establishing prognosis, monitoring treatment response, and guiding treatment selection [77]. In addition, based on their prognostic value in other human malignancies, CTCs were also investigated in neuroendocrine cancers, including PanNETs. Khan et al. were the first to analyze if CTCs can be detected in metastatic neuroendocrine cancers, whether they express epithelial cell adhesion molecules and predict radiological progression. Using the Cell Search platform, they analyzed blood samples from 79 patients with neuroendocrine tumors, including 19 PanNETs. CTCs were detected in 21% of PanNETs samples. CTCs levels correlated with urinary 5-hydroxy indole acetic acid (5-HIAA), liver metastases burden. There was a strong association between stable disease and the lack of CTCs [78]. The authors performed a second study using samples from 175 patients with neuroendocrine neoplasms; 42 patients had PanNETs. The presence of CTCs was linked to increased tumor burden, higher tumor grade, and elevated seric CgA. CTCs were also associated with worse PFS and OS, and they were able to define poor prognostic subgroups within the tumor grade groups [79]. To evaluate CTCs ability to predict treatment response, the same authors analyzed before and after treatment samples from 138 metastatic neuroendocrine neoplasm – 31 PanNETs. Patients without CTCs and those with a >50% reduction in CTCc count after treatment had better outcomes. CTCs variations were strongly correlated with OS (HR 4.13, P=0.0002) [80,81].
CTCs presence was also correlated with skeletal metastases. Rizzo et al. investigated samples from 254 patients, including 119 PanNETs; the authors demonstrated a significant association between CTCs presence and bone metastases (p < 0.0001). A cutoff value of ≥ 2 circulating tumor cells was able to identify patients with skeletal involvement in PanNETs. CTCs were also associated with tumor grade but not lung, lymph node, or peritoneal metastases [82].”
Thank you for the time spent analyzing our manuscript.

Reviewer 3 Report
The review by Havasi and colleagues aims at describing the implication of miRNA biology in PanNET.
The manuscript is well written, despite the paragraph should be correctly reorganized. Moreover the authors cover a very relevant topic on the biology of a such aggressive and deadly cancer.
Nevertheless, I have some important points to address:
- page 2 to 7: despite the given information, these pages described somewhat is out from the right topic of the manuscript
- by eliminating these pages, the core of the manuscript is too small and should be more described and discussed
Author Response
Manuscript ID:ijms-1539472
Title: “Current and new challenges in the management of pancreatic neuroendocrine tumors. The role of miRNAs-based approaches as new reliable biomarkers” by Havasi et al.
Reviewer 3 comments:
The review by Havasi and colleagues aims at describing the implication of miRNA biology in PanNET.
The manuscript is well written, despite the paragraph should be correctly reorganized. Moreover the authors cover a very relevant topic on the biology of a such aggressive and deadly cancer.
We want to thank the reviewer for the kind words and for the time spent reviewing our manuscript. We are confident that by answering the reviewer’s requests we will improve the current paper. Our main purpose is to present clear data which can be helpful for the clinicians and researchers involved in the field of gastrointestinal tumors.
Nevertheless, I have some important points to address:
- page 2 to 7: despite the given information, these pages described somewhat is out from the right topic of the manuscript
- by eliminating these pages, the core of the manuscript is too small and should be more described and discussed
We thank the reviewer for the comment. The main objective of this scoping review is to bring up-to-date information about PanNETs management with a special emphasis on miRNAs as possible reliable biomarkers. As medical oncologists and molecular biologists specialized in the field of gastro-oncology, we consider that the data available at the moment about the subject is scarce and the present review binds different specialities to better understand the subject. We consider that the background information about PanNETs is necessary for the general readership and also for the oncologists that might not be familiar with the field of gastrointestinal tumors. As a consequence, we propose this part to be further considered, to have complete data about what could mean the current and new challenges in the management of pancreatic neuroendocrine tumors
Thank you for taking your time and analyzing our manuscript.

Round 2
Reviewer 3 Report
I have no additional comments.
Great paper!